# Image-Based Malware Detection Using $\alpha$-Cuts and Binary Visualisation

**Betty Saridou** [1], **Isidoros Moulas** [2], **Stavros Shiaeles** [3,*] **and Basil Papadopoulos** [1]

1. Lab of Mathematics and Informatics (ISCE), Faculty of Mathematics, Programming and General Courses, Department of Civil Engineering, School of Engineering, Democritus University of Thrace, Kimmeria, 67100 Xanthi, Greece; dsaridou@civil.duth.gr (B.S.); papadob@civil.duth.gr (B.P.)
2. School of Computing, University of Portsmouth, Portsmouth PO1 2UP, UK; isidoros.moulas@port.ac.uk
3. Centre for Cybercrime and Economic Crime, University of Portsmouth, Portsmouth PO1 2UP, UK
* Correspondence: stavros.shiaeles@port.ac.uk

**Abstract:** Image conversion of malicious binaries, or binary visualisation, is a relevant approach in the security community. Recently, it has exceeded the role of a single-file malware analysis tool and has become a part of Intrusion Detection Systems (IDSs) thanks to the adoption of Convolutional Neural Networks (CNNs). However, there has been little effort toward image segmentation for the converted images. In this study, we propose a novel method that serves a dual purpose: (a) it enhances colour and pattern segmentation, and (b) it achieves a sparse representation of the images. According to this, we considered the R, G, and B colour values of each pixel as respective fuzzy sets. We then performed $\alpha$-cuts as a defuzzification method across all pixels of the image, which converted them to sparse matrices of 0s and 1s. Our method was tested on a variety of dataset sizes and evaluated according to the detection rates of hyperparameterised ResNet50 models. Our findings demonstrated that for larger datasets, sparse representations of intelligently coloured binary images can exceed the model performance of unprocessed ones, with 93.60% accuracy, 94.48% precision, 92.60% recall, and 93.53% f-score. This is the first time that $\alpha$-cuts were used in image processing and according to our results, we believe that they provide an important contribution to image processing for challenging datasets. Overall, it shows that it can become an integrated component of image-based IDS operations and other demanding real-time practices.

**Keywords:** image-based malware detection; binary visualisation; sparse matrix; alpha-cuts; fuzzy sets; defuzzification; image processing; convolutional neural networks; intrusion detection system; space-filling curves

## 1. Introduction

Image conversion of malicious binaries, or binary visualisation, is a relevant approach in malware detection and analysis. Recent advancements in Deep Learning (DL) and computer vision have allowed security researchers to successfully incorporate image processing techniques in their arsenal [1]. According to this approach, 2D visualisations of malicious and benign files can be used to train Machine Learning (ML)-based classifiers to detect the existence of malware in new entities. In this regard, binary visualisation is fast becoming a key component for Intrusion Detection Systems (IDSs) that wish to overcome the limitations of traditional signature-based techniques. For example, one such limitation is high signature generation sensitivity imposed by even minor differences in code among malware variants [2,3]. Another restriction that is forced by a signature-based detection scheme is the need for constant (and often manual) database updating [2].

Image-based IDSs will also play a vital role in the future as malware detection tools will have to comply with the EU's General Data Protection Regulation's (GDPR) "privacy by design" approach. According to this, malware detection software should avoid any personal data processing, or employ encryption/pseudoanonimisation techniques in any

other case [4,5]. For the moment, GDPR, which came into force in 2018, does not impose specific technical directions for applications that stand between security and personal data processing. At the same time, it releases any data that is processed to a state of anonymity from complying with the policy [5]. In that respect, image conversion of personal data (such as network traffic) can act as a form of masking user credentials and facilitate the development of GDPR-compliant security tools.

Image conversion of malware binaries is therefore a promising tool for the security community. Since the first attempts of visualising binaries [6,7] and the first image-based malware classification system [8], there has been a large volume of published studies experimenting with image generation or training classifiers. For example, some studies have been focused on texture analysis [8–10], or DL [2,3,11,12], all of which demonstrated positive results towards malware detection.

However, there has been little or no effort towards the investigation of image segmentation; that is, methods that could enhance the extraction of classes within an image. Since we are dealing with synthetic images, the elements we wish to specify are pixelated patterns that could reveal the presence of malicious activity. Two popular heuristics in image segmentation are colour and contrast, according to which, image segmentation techniques are able to identify shape boundaries based on pixel values. Still, these methods would prove difficult to implement in binary images, where information presentation is decided on an arbitrary colouring scheme and a highly sensitive sequence mapping.

As stated before, one of the goals of this study was to be able to work on byte-level data of files while preserving GDPR specifications. This would be violated, for example, when performing feature selection during static or dynamic analysis of malware files. For this reason, we chose a binary visualisation method that permits a level of anonymity to file contents while leveraging the advantages of optimal sequence mapping. The method has already been adopted by several research studies, including our own, showing good results in malware family classification [13–15], network traffic classification [16], and malware file detection [17].

Prior to this study, we had tested an image-based malware detection system where the visualisation colouring scheme was based on fuzzy sets. Continuing from that work, we now propose a novel method that enhances colour and pattern segmentation based on $\alpha$-cuts of the pixel colour values. The $\alpha$-cut is one of the most significant notions in fuzzy set theory for defuzzification, or the conversion of a fuzzy set to a crisp one. According to our method, we view the R, G, and B colour values of each pixel as respective fuzzy sets and perform an $\alpha$-cut across all pixels of the image. After the $\alpha$-cut, which acts essentially as a threshold, the new R, G, and B values for each pixel take either a value of 1 or 0 according to their initial value, producing new colour regions on the image. The collection of the processed images is then given to a Convolutional Neural Network (CNN), the ResNet50, to classify between malicious and benign images. For the purpose of this study, we explore the method of $\alpha$-cuts for $\alpha = 0.3$ and $\alpha = 0.6$ and compare the results to our previous method described in [17], as well as to [18], the original colouring scheme that inspired our work on binary visualisation.

In contrast to other image segmentation methods, our technique is not dependent on the colour space of the image. Usually, this is not the case with other segmentation methods, where practitioners have to find the colour space that will best serve their specific image segmentation problem [19,20]. This is even more evident in colour segmentation methods that involve partitioning the colour space (i.e., RGB or HSI) according to a thresholding method, e.g., measuring the Euclidean distance from the dominant colour [21,22]. $\alpha$-cuts for image processing are agnostic to the chosen colour space of an image and can be implemented as long as the pixel values belong in the $[0, 1]$ interval.

At the same time, our proposed methodology serves an additional role of computational importance. More specifically, it imposes a new form for the processed images that serve as input to the CNN. Indeed, after performing an $\alpha$-cut, the reduced $\alpha$-cut image is now comprised of a large number of 0 s, converting it to a sparse matrix. Interestingly,

because of the monotonically increasing shape of the fuzzy set we implemented in our method, the higher the value of $a$, the more sparsity the matrix exhibits. This could highly benefit the proposed IDS because of the computational advantages that are introduced when dealing with sparse representations of data in DL strategies. It is worth mentioning that the non-zero elements of the reduced images are 1 s, which highly contributes to the computational efficacy of matrix operations.

Even though we did not take any additional actions towards using the computational benefits of the sparse structure of the datasets, e.g., use specialised algorithms and data structures, our goal was to compare the results of different datasets of malicious and benign image representations using popular ML metrics. Our ultimate goal was to test the classification performance of malware/benign images that were processed using $\alpha$-cuts.

The rest of the paper is structured as follows: Section 2 gives an extensive review of the works in image-based malware detection, with a focus on the most recent innovations. Section 3 describes the proposed method of our image-based malware detection scheme that utilises fuzzy-based image processing. Section 4 presents the results, while Section 5 discusses the advantages and the limitations of our method. Finally, Section 6 concludes the paper.

## 2. Literature Review

### 2.1. Overview

The first to ever apply visualisation methods to malware files were [6,7]. Ref. [6] proved that self-organising maps were able to successfully group malware samples and enable easy identification of patterns in the data. Ref. [7] proposed the use of visualisation techniques to enhance the reverse engineering process and help analysts understand and interpret the structure and contents of the binary. However, in 2011, Ref. [8] presented the first malware classification system by comparing binary texture analysis and dynamic analysis. In their work, binary texture analysis slightly outperformed dynamic analysis, although dynamic analysis was able to provide additional information on the behaviour of the malware. Since then, the field of image-based malware classification has attracted significant attention within the security community, as it constitutes a promising solution for detecting malware in a fast and accurate manner. Recent methodologies range from traditional ML algorithms to more advanced DL models. In this paper, we present the newest additions in the visualisation-based malware detection domain and we divide them into three categories: (a) malware/benign file detection, described in Section 2.2; (b) malware file detection and family classification, described in Section 2.3; and (c) network traffic malware detection/classification, described in Section 2.4. We also discuss their most common limitations in Section 2.5.

### 2.2. Malware/Benign File Detection

In their paper, Ref. [17] proposed a system based on binary visualisation with a fuzzy set-based colouring scheme. The images were then passed to several CNN classifiers to detect malicious behaviour. Ref. [23] presented a pattern-based approach for insider threat classification, where they used image-based feature representation to convert log data into images for Wavelet CNN classification. Ref. [24] proposed a Windows malware detection system using CNN and AlexNet learning models. Image conversion of Windows PE files was also the subject of [25], who performed an extensive experiment of a malware detection technique based on RGB images. More specifically, they tested 15 fine-tuned DL models for feature extraction and 12 ML algorithms as the final classifier, from which the winning models were RegNetY320 and SVM. In their research, they also employed data augmentation and transfer learning. Even though [26] did not achieve high accuracy scores when applying CNN classification to their visualisations, they presented an analysis of the interpretability of the malicious patterns within the images. Ref. [27] proposed a visualisable malware detection method based on multi-dimension dynamic behaviours. However, their analysis was carried out on a limited malware dataset. Ref. [28] introduced

an ensemble architecture for malware detection that is adaptable to different types of malware. The authors developed a combination of multiple ML algorithms and used an ensemble technique to improve the accuracy of the model. Ref. [29] also published malware detection using deep CNN in combination with image-based representations of malware samples.

### 2.3. Malware Detection and Family Classification

In their work, Ref. [15] proposed a hybrid framework for image-based malware classification based on space-filling curves. The authors argued that their approach could effectively capture the relationships between binary code and its associated graphical representation. Their approach used traditional ML algorithms to achieve high accuracy and efficiency, outperforming existing malware classification methods. Ref. [2] presented an enhancement for image-based malware classification using ML with low-dimension normalised input images to reduce the dimensionality of the data. However, as they mentioned, the normalisation process may not be suitable for all types of malware. Ref. [30], who had previously used CNN for malware feature extraction in [31], incorporated fuzzy logic in their approach by using convolutional fuzzy neural networks based on feature fusion and the Taguchi method. Ref. [32] presented a malware detection model that is based on Multi-Feature Fusion (MFF) and Histogram of Dynamic Binary Analysis (HDBA). The authors experimentally demonstrated its ability to detect malware variants and its high accuracy in comparison to existing methods with a detection time of 9.63 ms.

To compensate for "the problem of unpredictable truncation" caused by the use of different image widths in malware visualisation, Ref. [33] proposed a malware classification method based on a fusion of Efficient-Net and 1D-CNN. Their method exhibited high accuracy in detecting malware and the ability to classify different malware families. Ref. [34] introduced S-DCNN, an effective DL model which used image representation of binaries and combined transfer and ensemble learning. The model consisted of three CNNs (ResNet50, Xception, and EfficientNet-B4) and a multilayered perceptron as the final classifier. In their work, Ref. [35] presented a triplet neural network (NN) approach to learning similarities between vectors in the latent space. Their method used evolutionary optimisation and an ensemble network to return results by separating entangled representations. The technique demonstrated the segregation of Trojan-based structural features and demonstrated the lower segregation of Simda-typed data in the latent space. Ref. [36], who employed Generative Adversarial Networks (GANs) in their experiment, presented Iron-Dome, a multi-modal malware detection system. Iron-Dome secures IoT networked systems at runtime by classifying executable-based greyscale images.

The authors in [37,38] proposed a new method for malware detection that combined Markov images and transfer learning. The experiments were successful in terms of accuracy and speed but at the same time, they may require more computational resources compared to traditional ML approaches. Ref. [39] proposed the use of visualisation techniques along with the B2IMG and Gabor Filters as image augmentation methods and various CNN architectures to improve malware detection results. Their results were optimal for the B2IMG technique when combined with coloured images. Ref. [40] discussed the use of transfer learning in a EfficientNet3 CNN to classify malware efficiently. The authors utilised pre-trained deep NNs as a base model and fine-tuned them on malware data to achieve high accuracy. According to them, transfer learning proved that it can significantly reduce the training time and computational resources required. Ref. [41] concentrated on reducing the effort needed for malware data labelling. The authors suggested a technique for choosing a subset of data to serve as prototypes for representing the complete dataset and recommended the use of VGG16.

Ref. [42] presented a new approach to malware detection using greyscale binary visualisation and a hyperparameterised CNN. Their method achieved a high accuracy score, while they also provided the code for the training process. Ref. [43] proposed a multilayer DL approach for malware classification in 5G-enabled Industrial Internet of

Things (IIoT) systems. The approach used a combination of CNN to detect malware, even in 5G systems, where malware attacks are becoming increasingly common, offering real-time connectivity. Ref. [44] focused on Windows malware detection by exploring the effectiveness and efficiency of the LightGBM algorithm, a gradient-boosting framework that uses tree-based learning algorithms. The authors claim that their approach was effective and fast, both in detecting and classifying malware. In their article, Ref. [45] experimented with various colour models in their visualisation technique to perform malware classification methods. All colour model techniques exhibited equally high performance when they were paired with feature extraction and the SVM model. Ref. [46] presented BinImg2Vec a method that transformed the binary code of malware samples into images and then used the data2vec framework to embed the images in a low-dimensional space. The embedded images were then used as inputs to a CNN for classification.

Ref. [47] proposed MalCNN, a new enhancement for malicious image classification using NNs. Even though the authors claim that their approach outperformed existing methods in terms of accuracy and efficiency, they did not provide a comprehensive comparison with other methods to validate this claim. Additionally, the article did not mention any limitations or potential drawbacks of the proposed method. Ref. [48] presented a static malware classification approach using greyscale image representation and lightweight CNNs that is suitable for IoT environments. Ref. [49] proposed a new method using a combination of automated transmutation and CNNs. However, these articles did not provide a detailed comparison of their proposed methods with other existing methods. Ref. [50] presented a lightweight CNN for image-based malware classification on embedded systems. The authors claim that their approach is effective in classifying malware on embedded systems because of its limited demands on computational resources. Ref. [51] used a Multi-Layer Perceptron (MLP) model. The authors evaluated their method on a dataset of malware images to show the efficiency of MLP, which was highly hyperparameterised. Ref. [52] attempted a malware classification method based on a lightweight architecture of CNN called MalShuffleNet, which is designed to reduce the computational cost of the model without sacrificing accuracy.

There were also several studies that incorporated the use of GANs in their methodology. Ref. [53] converted malware executables into images, which were then used to train a GAN for family classification. The authors used a variety of techniques for image generation and ML classification. The results favoured the combination of colourmap images and AC-GAN. Ref. [54] used Auxiliary-classifier GAN (AC-GAN) to generate malware images and then evaluate them. The authors found that fake malware images did not impose a threat to adversarial attacks; however, the models were able to successfully classify generated and real malware images. In their study, Ref. [55] used an augmentation model to boost the classification results of malicious images based on their method B1IMG. The CNN-based achieved superior results using RGB photos as opposed to grey images. Ref. [56] also used GANs to enrich samples from the same family and performed classification with CNN. Ref. [57] presented an analysis of the robustness of image-based malware analysis. More specifically, they compared gist descriptors to CNN that were trained directly on malware images to test their malware obfuscation resistance. According to their experiment, they discovered that gist descriptors were more reliable than CNNs.

### 2.4. Network Traffic Malware Detection/Classification

In their work, Ref. [58] proposed a DL-based model for malware traffic classification using PCAP (packet capture) data. The model converts PCAP data into images and uses image-based NN models to classify the malware traffic. The authors used vision transformers and CNN. Ref. [59] proposed a method for detecting IIoT malware. More specifically, they proposed an edge computing-based malware detection system that identifies malware by sending massive amounts of IIoT traffic data from smart factories to edge servers. By extracting relevant features from the network traffic images, they used DL techniques to identify malicious behaviour. Ref. [60] proposed an ML-based intrusion detection and

response system. The system focused on network profiling and trained ML algorithms on historical attack data to detect intrusions in real time. Ref. [61] proposed a method that framed network flows as images and used image recognition algorithms to detect anomalies. The authors used federated learning to train the algorithms and evaluated the performance of their system on DDoS attack data.

### 2.5. Limitations of Existing Methods

These articles provide different approaches to tackle the problem of detecting and classifying malicious activity based on visualisation methods. While these techniques show promising results, they also present certain limitations:

- Limited dataset or limited diversity in the dataset used for training and testing the models, which may limit the generalisability of the models to real-world scenarios.
- Some studies have reported high accuracy rates, but the models may have to overfit to the training data and may not perform as well on unseen data.
- Use of only static images of malware samples, which may not capture the dynamic behaviour of malware.
- Vulnerability to adversarial attacks that can evade the detection of the models.
- The computational complexity of some models may make them unsuitable for deployment on real-world scenarios with low-power and resource-constrained devices.
- Lack of interpretability of some models, which may make it difficult to understand how they arrive at their classifications and limit their adoption in security-critical applications.
- Collecting and labelling malware samples is a labour-intensive and time-consuming process, and some studies have proposed techniques to reduce labelling efforts. However, the effectiveness of these techniques remains to be evaluated.

### 3. Materials and Methods

#### 3.1. Overview

In this study, we propose an image processing method and we test it on binary visualisation images as part of a CNN-based malware detection mechanism. More specifically, clean and malicious files are converted to pixelated images by using design and chromatic schemes that successfully capture patterns of byte-level data. Next, an image processing technique based on a defuzzification method is applied to the initial image dataset. This step is aimed at the development of new image datasets that may enhance (or isolate) malware-indicative patterns during detection. Finally, we train a CNN model for all labelled image datasets to assess their overall performance. Our proposed methodology is a continuation of our previous work on binary visualisation and malware detection, where we tested a fuzzy-based colouring scheme instead of the traditionally used solid colours.

#### 3.2. Clean and Malicious File Collection

The first step in developing the method was the acquisition of a variety of files with verified clean and malicious content. In this regard, the collection of clean and malicious files originated from the *Contagio* dataset, a malware dump that provides sorted positives and negatives for security researcher purposes. More precisely, *Contagio* offers a total of 16,800 clean and 11,960 malicious files with a predominance of the PDF file format. Other file types included ELF, JAR, EXE, MACH-O, XLS, DOC, and RTF, some of which had been through the process of encryption. This curated dataset was a result of various other open source datasets and is made available in [62].

#### 3.3. Creation of Image Datasets

3.3.1. File Conversion to Image Format

On completion of file acquisition, we proceeded with converting the files to images. As previously stated, the goal of this method was to create byte-level visualisations of the file structure that could unveil the presence of obfuscated code. This requires a layout to map the one-dimensional sequence of ASCII characters in the 2D space, along with an

intelligent colouring scheme for byte representation. The information we are dealing with is in the form of a byte sequence. Therefore, it is indispensable to take advantage of all possible information that can be derived from it. This proves to be a challenging task in information representation, especially when it is combined with data transformation from a low to a higher-dimensional space.

Regarding image dimensions, we needed to acquire fixed-size images that would be appropriate as input to CNNs. In general, CNNs require images of identical size; therefore, we decided to use 256 × 256 images, a common image size for CNN training. Moreover, this particular size was useful because it represents a medium-value image; given the difference in size among malware (and benign) files, a single file that was large enough would correspond to multiple converted images. Hence, small sizes were unsuitable in our methodology, but at the same time, a larger size could negatively affect training time and performance. It should also be noted that after the byte sequence was fully mapped, any remaining empty space of the converted 256 × 256 image was filled with black pixels.

In Sections 3.3.2–3.3.4 of this section, we start by introducing key concepts that were involved in the construction of the files into 2D representations. These ideas include colour assignment, colour tone adjustment, and mapping layout. In Figures 1 and 2, we provide examples of a malicious and a benign file, respectively, that were converted to 2D representations using these concepts.

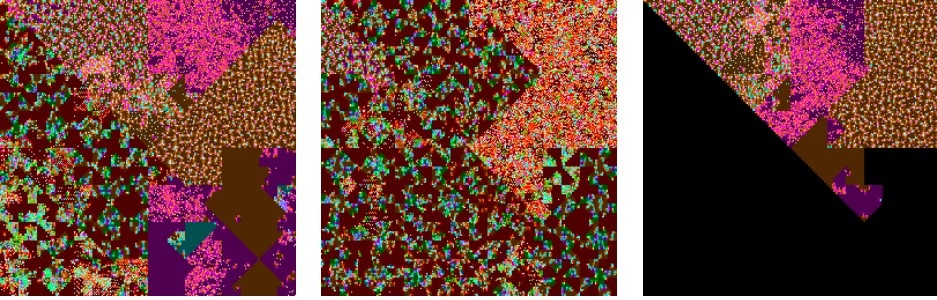

**Figure 1.** A 2D representation of a malicious .rtf file (159.71 KB) with the `msword/cve20103333` Trojan.

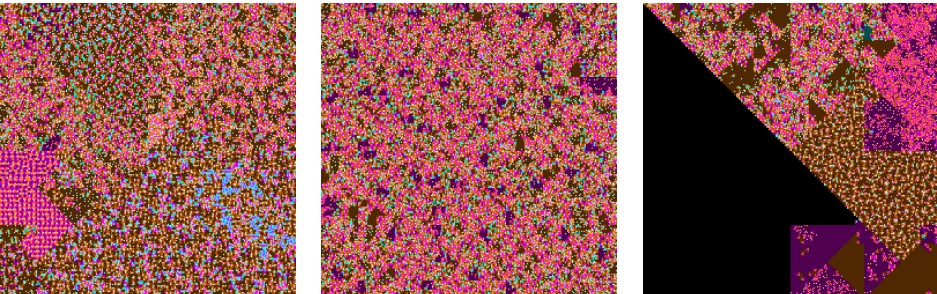

**Figure 2.** A 2D representation of a clean .rtf file (175 KB).

### 3.3.2. Colour Assignment per ASCII Character Class
### The RGB Colour Model

In computer graphics and digital image processing, the RGB colour model is the most popular model for colour representation today [63,64], while its development has its origins in James Maxwell's work in the 1860s, on the nature and the perception of colour [65,66]. The RGB is an additive colour model, which means that each possible colour value and tone can be created by adding combinations of the red, green, and blue primary colours of light. More precisely, RGB stands for the Red, Green, and Blue values, and all of them fall in the [0, 255] or [0, 1] interval [63,64]. As a consequence, each pixel is represented by a 3-tuple that includes the R, G, and B channel values.

ASCII Colour Classes

According to our previous work in [17], the ASCII table was divided into 5 groups, each to be represented by a characteristic colour. In this paper, based on empirical malware analysis of files, the further class division was performed in order to extend pattern recognition. As a result, we expand the division groups into 8 different categories. The 8 colour groups represent the following character types: (1) magenta for letters (uppercase and lowercase) which are a subset of the printable characters; (2) orange for numbers which are a subset of the printable characters; (3) blue for space which is a subset of the printable characters; (4) cyan for the remaining printable characters; (5) green for control characters; (6) red for extended characters; (7) grey for the null character; and (8) yellow for the non-breaking space. Table 1 provides a summary of the various colour classes.

**Table 1.** The colours and their corresponding RGB values for the 7 categories of the ASCII characters.

| Character Type | ASCII Decimal Value | Colour Class | RGB Values |
| --- | --- | --- | --- |
| control | 1–31, 127 | green | (0, 1, 0) |
| extended | 128–254 | red | (1, 0, 0) |
| the NULL character | 0 | grey | (0.5, 0.5, 0.5) |
| the non-breaking space | 255 | yellow | (1, 1, 0) |
| numbers (part of printable) | 48–57 | orange | (1, 0.5, 0) |
| letters (part of printable) | 65–90, 97–122 | magenta | (1, 0, 1) |
| space (part of printable) | 32 | blue | (0, 0, 1) |
| remaining printable | 33–47, 58–64, 91–96, 123–126 | cyan | (0, 1, 1) |

### 3.3.3. Colour Tone Adjustment through Fuzzy Set Theory

Fuzzy Sets

A *fuzzy set* $\tilde{A}$ is a generalised type of a classical set where given a *universal set X*, the elements of $X$ are assigned a value from the unit interval $[0,1]$ denoting their degree of membership to $\tilde{A}$. The characteristic function of each fuzzy set that assigns values within this range is called a *membership function* $\mu_{\tilde{A}}$:

$$\mu_{\tilde{A}} : X \to [0,1].$$

with larger values expressing a stronger membership to $\tilde{A}$.

One of the main advantages of fuzzy sets is the ability to express vague ideas in simple linguistic terms [67]. In our work, we wished to use this property to express the concept of colour tone of a pixel as a byproduct of its immediate environment. We continue from our previous work in [17], where we defined a Fuzzy Inference System (FIS) to decide on the lightness (or darkness) of each pixel, depending on its similarity with its neighbouring pixels to its left and to its right side. The FIS comprised two inputs and one output. The inputs denote the colour similarity to the left and the colour similarity to the right pixels, respectively, while the output denotes the adjustment in colour tone for the pixel in question. According to the FIS, the colour of the pixel we are dealing with turns darker if neighbouring pixels from both sides share the same colour (according to a certain degree), retains a neutral tone if only *some* of the neighbouring pixels share the same colour (according to a certain degree), and turns whiter if neighbouring pixels are dissimilar to it altogether. In Table 2, we provide a summary of the FIS, while in Table 3 we provide the fuzzy rules. For more details on the construction of the FIS, the reader is advised to read our work in [17].

**Table 2.** A summary of the FIS that adjusted the colour tone of each pixel and we first implemented in [17].

| FIS | |
|---|---|
| **Type** | Mamdani |
| **Inputs** | Left Similarity, Right Similarity |
| **Ouput** | Colour Tone |
| **Implication** | min |
| **Aggregation** | max |
| **Defuzzification** | centroid |

**Table 3.** The rules of the FIS. The neighbouring similarity of the pixels to the left and the right affects the colour tone.

| Rule | Left Similarity | Right Similarity | Colour Tone |
|---|---|---|---|
| 1 | Different | - | Light |
| 2 | Similar | - | Medium |
| 3 | Same | - | Dark |
| 4 | - | Different | Light |
| 5 | - | Similar | Medium |
| 6 | - | Same | Dark |

3.3.4. The Mapping

The H-Index Space-Filling Curve

In mathematical analysis, a Space-Filling Curve (SFC) is a continuous curve that "maps a one-dimensional space to a higher dimensional space" [68]. The first SFC was discovered by Giuseppe Peano in 1890, who was inspired by Cantor's work on the infinite number of elements in the unit interval. More specifically, SFCs, which are considered to be special cases of fractals, formed the basis for Mandelbrot's work in [69]. Ever since, more SFCs have been discovered, with Hilbert's curve being by far the most popular one in engineering applications. Some of these applications include signal processing [70], computer graphics [71], and cloud storage [72].

In our previous work, two SFCs were tested for CNN-aided malware detection, namely the Hilbert curve, discovered by [73], and the H-indexing (or H-order, or simply the H-curve), discovered by [74]. Out of these two curves, the H-curve provides a better locality property. In other words, points that are close together in the one-dimensional space, tend to stay close together when mapped in a higher dimension. For this reason, the H-curve was chosen to be the mesh indexing of our experiments. Moreover, for the purpose of this study, we used 256 × 256 images, or the seventh iteration of the H-curve mapping. It is worth noting that the H-curve is a Hamiltonian curve, which in our method starts and ends on the top left of the image. Figure 3 shows the third iteration of the H-curve mapping.

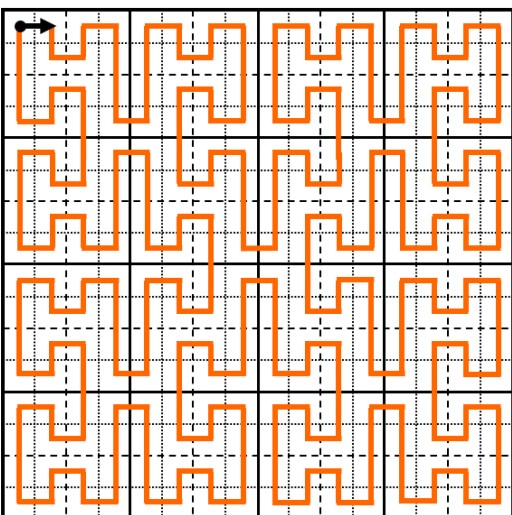

**Figure 3.** The H-curve, a Hamiltonian curve, was discovered by [74] in 1997. This is the sequential order for a $16 \times 16$ representation.

### 3.4. Generating More Image Datasets Based on $\alpha$-Cuts.

3.4.1. Creating Crisp Sets with an $\alpha$-Cut

The $\alpha$-Cut

One of the most important concepts of fuzzy sets is the $\alpha$-cut: According to [67], given a fuzzy set $\tilde{A}$ defined on the universal set $X$ and any number $\alpha \in [0,1]$, the $\alpha$-cut $^{\alpha}A$ is the crisp set

$$^{\alpha}A = \{x \mid \mu_{\tilde{A}}(x) \geq \alpha\}.$$

More precisely, the $\alpha$-cut of a fuzzy set $\tilde{A}$ is the classical set $^{\alpha}A$ that contains all the elements of $X$ that belong to $\tilde{A}$ with a membership degree that is greater than or equal to the value $\alpha$. To put it differently, any element $x \in {}^{\alpha}A$ is also an element of $\tilde{A}$ with a degree of membership that is greater or equal to the $\alpha$ value [75]. It is worth noting that, in the literature, $\alpha$-cuts are sometimes called $\lambda$-cuts (or *lambda* cuts).

In Section 3.3, we fully addressed the process of converting files into images. Next, we incorporated the concept of $\alpha$-cuts on that first image dataset to create more image datasets that could potentially improve detection. It is worth noting that this technique was implemented on the finished image dataset, or, in other words, after each pixel had its colour tone adjusted.

To achieve this, we treated the RGB colour channels of each image as fuzzy sets and reduced them into $\alpha$-cut sets. More specifically, we considered each colour (red, green, and blue) to be a fuzzy set, where each colour value denoted the degree of membership to this particular set. The shape of the fuzzy colour set, which is identical for all three colours, is displayed in Figure 4, along with an $\alpha$-cut as an example.

According to this shape, the red, green, and blue fuzzy sets were reduced to an $\alpha$-cut of the form $^{\alpha}A = [a, 1]$. Practically, this meant that any values (for the red, green, or blue fuzzy set, respectively) that were greater than or equal to $\alpha$ became equal to 1 and the rest became equal to 0. We provide an example of this practice in Figure 5 for $\alpha = 0.3$.

At the pixel level, in terms of colour theory, reducing images to their $\alpha$-cuts translates into a sparser representation of colour in the image. For example, after taking an $\alpha$-cut of an image with an $\alpha = 0.3$, a pixel with previously RGB values of [0.35, 0.17, 0.58] would become [1, 0, 1]. Naturally, images that are $\alpha$-cuts of other images are comprised of only 8 solid colours, namely, black ([0, 0, 0]), white ([1, 1, 1]), red ([1, 0, 0]), green ([0, 1, 0]), blue ([0, 0, 1]), yellow ([1, 1, 0]), magenta ([1, 0, 1]), and cyan ([0, 1, 1]). In other words, they include all possible combinations of 0 and 1 in the RGB colour model ($2^3 = 8$). Essentially, this whole process generates a new image, as the new values per pixel create new colourings.

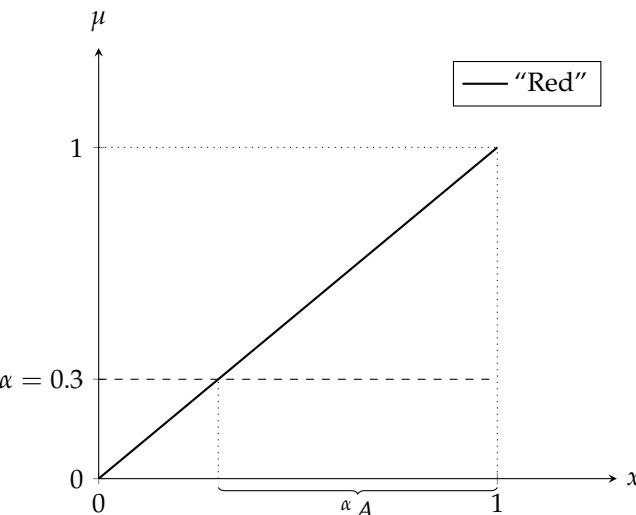

**Figure 4.** The fuzzy set for the value *colour*. Here, we display the colour *Red*, although the same shape applies for *Green* and *Blue*. We also demonstrate the crisp set $^{0.3}A$.

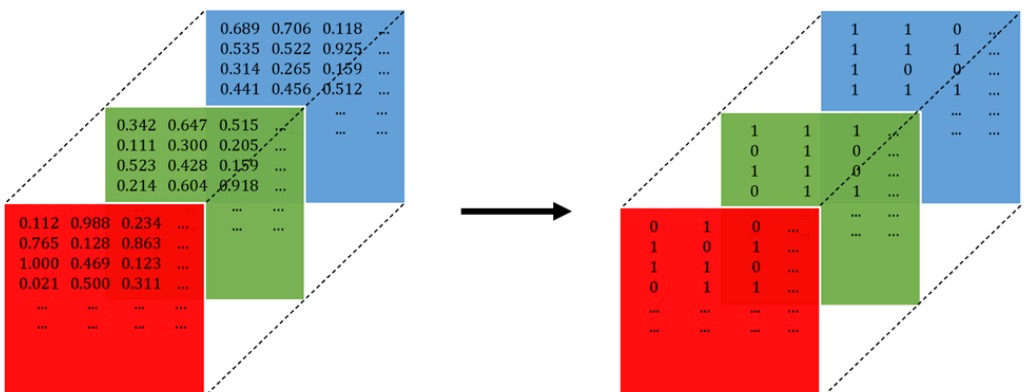

**Figure 5.** We considered each R, G, and B colour array of an image to be a fuzzy set (**left**). We then reduced each set to an $\alpha$-cut (**right**) according to a value $\alpha$. In this example, $\alpha = 0.3$. The newly created $^{\alpha}A$ sets were populated according to this: any element of each fuzzy set that was greater than or equal to 0.3 was also an element of the crisp set $^{0.3}A$ and therefore took a value of 1, with the rest taking a value of 0.

In our study, we tried several $\alpha$ values on images from the initial dataset. More specifically, we extracted $\alpha$-cuts for $\alpha = \{0.1, 0.2, 0.3, 0.4, 0.5, 0.6, 0.7, 0.8, 0.9\}$. According to the shape we chose to represent the fuzzy set colour (see Figure 4), it was interesting to observe the graduate darkening of colours as we increase $\alpha$. However, for our experiments, due to large running times, we run our experiments for the original unprocessed image dataset and reduced image datasets that resulted from $\alpha = 0.3$ and $\alpha = 0.6$. In Figure 6, we display a malicious image and the various $\alpha$-cut images from 0.1 to 0.9. We also display the same processing to a natural image (a non-synthetic one) in Figure 7, to better convey the concept of colour segmentation.

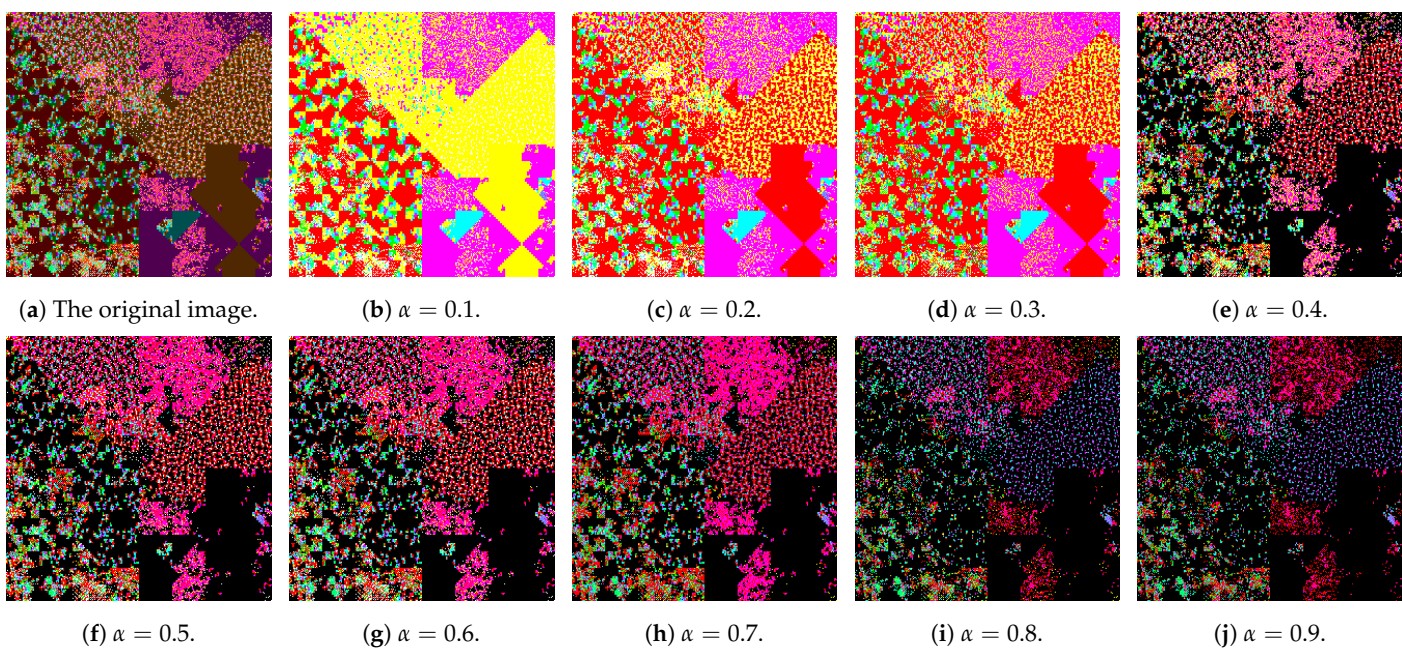

**Figure 6.** A malicious image in (**a**) and reduced images of it, for $\alpha = [0.1, \ldots, 0.9]$ in (**b**–**j**), respectively.

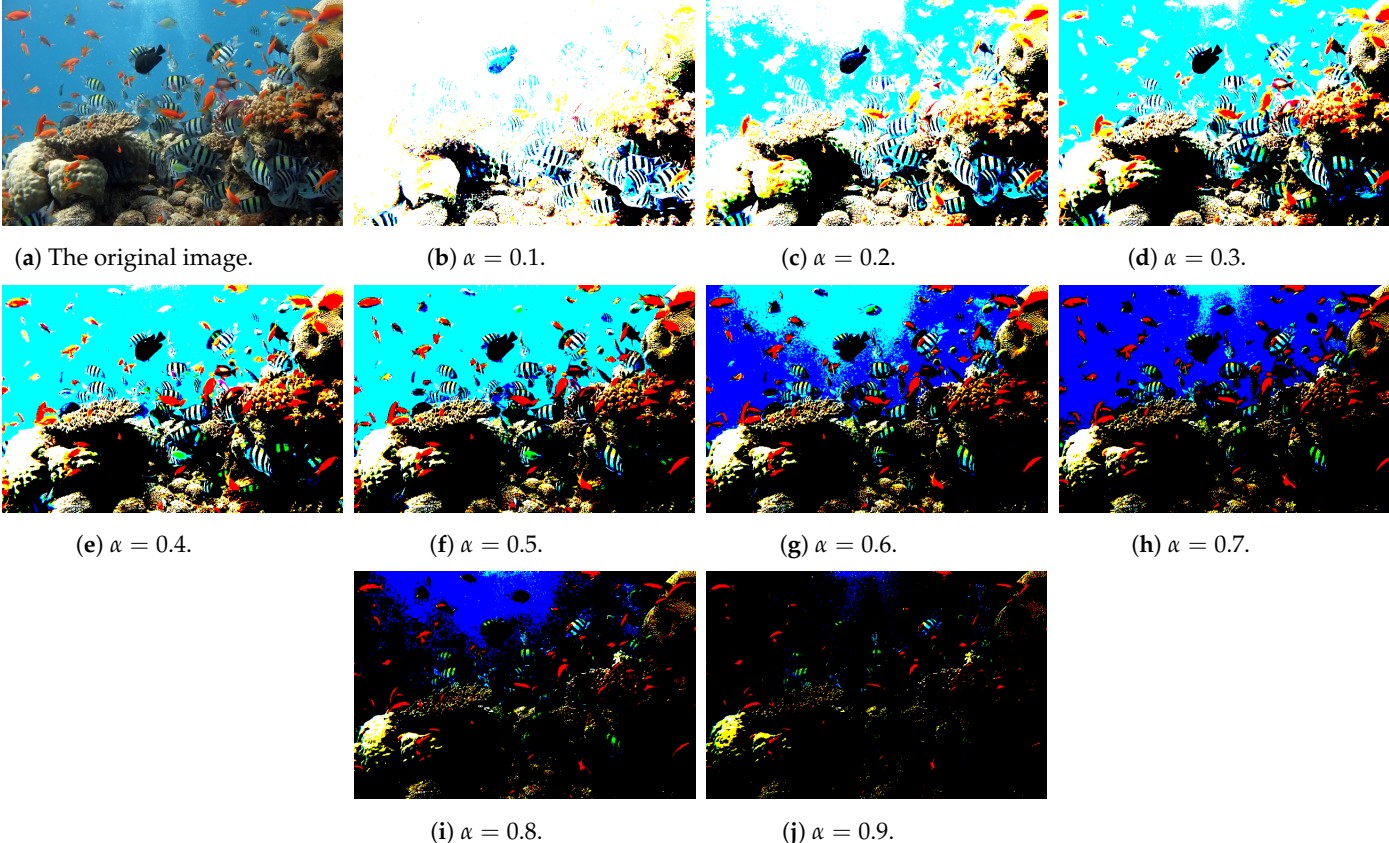

**Figure 7.** A natural image in (**a**) and reduced images of it, for $\alpha = [0.1, \ldots, 0.9]$ in (**b**–**j**), respectively.

### 3.5. Deep Learning for Image Recognition

Convolutional Neural Networks

CNNs are a class of deep neural networks that were developed in the 1980s and drew their inspiration from research on the visual cortex of the brain [76–82]. Ever since, they have been employed in image recognition, and in 1998, they had their major breakthrough when a series of publications by [83–87] led to the recognition of handwritten digits. As

traditional artificial neural networks would fail in image pattern recognition (given their fully connected layers and a large number of pixels per image), CNNs provide a solution by deploying partially connected layers and weight sharing. This cost-effective strategy is achieved by introducing two novel building blocks: the convolutional layer and the pooling layer [88]. Typically, CNNs are trained using supervised learning techniques and tested on a testing dataset they have never seen before. Currently, there are multiple sophisticated CNN architectures (with many variants) that can successfully classify millions of images into several thousands of classes [89]. In this paper, to successfully detect patterns of malicious code in file-based images, we deploy the ResNet50 architecture.

The ResNet50 is a variant of the Residual Network or ResNet architecture. ResNet was introduced by [90], when he won the ILSVRC challenge in 2015, with the original model composed of 152 layers. Other variants include ResNet18, ResNet24, and ResNet101. The ResNet architecture relies on a stack of residual units, where smaller neural networks with skip connections pass the signal to the whole network. The residual connections allow for the network to learn a more efficient representation of the data. This helps to reduce the vanishing gradient problem that occurs in deeper networks, making it easier to train deep models. Its depth allows the network to learn complex features and patterns in the data, leading to better classification performance. However, despite its deep architecture, ResNet50's functionality speeds up training significantly and results in a total of 23 million trainable parameters. From previous work on binary visualisation images in [91], ResNet50 was shown to generalise well to new and unseen images, making it a reliable choice for classification tasks of synthetic images. In general, it offers several advantages over other models, including improved training, better performance, and generalisation. It has a total of 48 Convolutional layers, 1 Max Pooling layer, and 1 Average Pooling layer.

It is important to note that each ResNet50 model of our experiment was implemented from scratch without any use of transfer learning. Our models were hyperparameterised for batch size, optimiser, and learning rate. We also employed early stopping, and since we are dealing with binary classification, our activation function of choice was the sigmoid function. Finally, for each dataset, we used 80% of the images as the training dataset, 10% as the validation dataset, and 10% as the testing dataset.

### 3.6. Performance Evaluation Metrics

To evaluate the performance of our proposed methodology, we employ the Accuracy, Precision, Recall, and F-score metrics, all of which are based on the following measures: True Positive ($TP$), True Negative ($TN$), False Positive ($FP$), and False Negative ($FN$).

$TP$ is the number of all elements of a class that were correctly predicted to belong in that class or the number of malicious images that were correctly identified as malicious. $TN$ is the number of all elements outside of a class that were correctly predicted to belong in a different class or the number of benign images that were correctly identified as benign. $FP$ is the number of all elements outside of a class that were incorrectly predicted to belong in that class or the number of benign images that were incorrectly identified as malicious. $FN$ is the number of all elements of a class that were incorrectly predicted to belong in a different class, or it is the number of malicious images that were incorrectly identified as benign. According to these definitions, we describe the following metrics:

- Accuracy is the percentage of images (malicious and benign) that were correctly predicted:

$$Accuracy = \frac{TP + TN}{TP + TN + FP + FN} \tag{1}$$

- Precision is the percentage of correctly predicted malicious images over the total amount of predicted malicious images:

$$Precision = \frac{TP}{TP + FP} \tag{2}$$

- Recall is the percentage of correctly predicted malicious images over the total amount of malicious images:

$$Recall = \frac{TP}{TP + FN} \tag{3}$$

- F-score is the harmonic mean between Precision and Recall, and demonstrates the robustness of a model:

$$F\text{-}score = \frac{2TP}{2TP + FP + FN} \tag{4}$$

To assess model performance, we also plotted the learning curves of the training and validation datasets. Learning curves are a commonly used visual diagnostic tool in DL that indicate the performance of the model over time. More specifically, they provide insight into the adequacy of the datasets and the model's ability to generalise: after enough experience, two pairs of curves, the training and validation accuracies, and the training and validation losses should follow an increasing and a decreasing pattern, respectively, in a stable manner. The model is shown to be a good fit when the training and validation loss curves lower until they reach a stable value, with the latter being slightly higher than the former, or when the training and validation accuracy curves increase until they reach a stable value, with the latter being slightly lower than the former.

### 3.7. Programming Language

The prototype of this work was written initially in Python, a high-level general-purpose language that is suitable for scientific programming [92]. Python was used throughout the research process, from image conversion and fuzzy image processing to deep neural network training and classification. Because of the scalability of our work, image conversion, image processing, and initial CNN training was built locally, using the Anaconda Distribution 2.3.2 that employs Python 3.7. More specifically, we used both the Spyder IDE 5.1.5 and Jupyter Notebook 6.5.2. Notably, the fuzzy inference system for adapting the colour lightness for each pixel was built with the `scikit-fuzzy 0.4.2` (or `skfuzzy`) Python package by [93], while image processing was built with the `pillow 9.2.0` and `numpy 1.21.5` packages. Initial CNN training relied on `keras 2.3.1`. However, when computational needs for CNN training grew, we relied on a Python 3.8 environment, that employed `keras 2.9.0` and `scikeras 0.10.0`.

### 3.8. GPU Resources

Image conversion and processing were initially executed on local CPU resources of an Intel Core i7-7500U at 2.7 GHz and 16 GB RAM; however, CNN training required more computing power. To speed up training times, we used Google Colab's subscription services [94]. Google Colab is a cloud-based service that provides free interactive coding environments that allow users to quickly develop and run code without the need to install software on their own computers. It offers a Jupyter Notebook-like environment [95] and can be configured to use GPU (or even TPU) resources with up to 128 GB of RAM and 8000 Tensor Cores.

## 4. Results
### 4.1. Overview

In this section, we present the results of our experiments. To successfully determine the effectiveness of our proposed methodology, we compared it to [18] (the colouring method that inspired this work) and to [17] (our previous work and the evolution of [18]). The colouring schemes were adapted to our own image conversion methodology, according to which a single binary file outputs multiple images as per its size. This was deemed necessary in order to solely compare the effectiveness of the colouring methods and eliminate the computational advantages of a single output image. In other words, when testing the visualisation method of [17,18], we produced multiple images from a single mali-

cious/benign file (as we would normally do in our proposed method), but followed the colour assignment method that was presented in their work.

*4.2. Experimental Results*

According to our methodology, we evaluated our images in terms of the successful colour-based pattern construction of ASCII characters, but as described in Section 3.4, we also tested reduced datasets of them, namely for $\alpha = 0.3$ and $\alpha = 0.6$. Additionally, we implemented this image processing technique for the colouring methodology of [17] to achieve a complete comparison between the two methods. Notably, the works of [17,18] performed limited clustering of the ASCII Table; hence, these images exhibited fewer patterns compared to ours. Moreover, we acknowledge the importance of dataset size in ML procedures and tested the various methods for four different sizes, namely 1250, 2500, 5000, and 10,000 images. Each dataset consisted of 50% malicious and 50% benign images. Overall, we ran a total of 28 experiments in order to include all different combinations of colour class methodologies, $\alpha$-cut processing, and dataset size.

A comparison of the various methods is presented in Tables 4–10. In Tables 8–10 we show the results of our proposed method. Table 8 describes the evaluation scores of the curated image datasets with the colour classes described in Table 1, while Tables 9 and 10 present the results of the reduced image datasets for $\alpha = 0.3$ and $\alpha = 0.6$, respectively. As is demonstrated, an increased dataset size has a positive effect on the training of ResNet50. More specifically, for the experiments with unprocessed images (without taking an $\alpha$-cut), the dataset of 1250 images delivered 86.40% accuracy, 92.59% precision, 79.36% recall, and 85.47% F-score, while the dataset of 10,000 delivered 93.89% accuracy, 96.40% precision, 91.20% recall, and 93.73% F-score. As it can be observed, these improved scores for the largest dataset seem to perform slightly worse than their counterparts for the unprocessed images of the [17] methodology in Table 5, which delivered 94.50% accuracy, 97.23% precision, 91.60% recall, and 94.33% F-score.

Evaluation metrics for the model of the reduced dataset for $\alpha = 0.3$ in Table 9 show that it does not perform better than the model for the unprocessed dataset in Table 8. Indeed, its best-performing model delivered 93.40% accuracy, 96.56% precision, 90.00% recall, and 93.16% F-score. The same notion is repeated with the reduced dataset of $\alpha = 0.6$ in Table 10, where the best-performing model delivered 93.60% accuracy, 94.48% precision, 92.60% recall, and 93.53% F-score.

We also observed similar results for the reduced datasets of the colouring method of [17] in Tables 6 and 7. According to them, the best performing model of the reduced dataset with $\alpha = 0.3$ delivered 93.89% accuracy, 95.82% precision, 91.80% recall, and 93.76% F-score, while the best performing model of the reduced dataset of $\alpha = 0.6$ delivered 92.80% accuracy, 95.72% precision, 89.60% recall, and 92.56% F-score.

From these results, we conclude that reducing images with the help of $\alpha$-cuts did not increase the model's accuracy metrics. However, it is worth mentioning that the differences between the scores of each method were numerically insignificant. In any case, the best performing models of both the unprocessed and the reduced datasets for our proposed method and the work of [17] significantly surpassed the best performing model of [18] in Table 4, which reported 89.80% accuracy, 92.70% precision, 86.40% recall, and 89.44% F-score.

Despite the seemingly unimproved metrics we acquired when applying the $\alpha$-cuts to our dataset, we observed a significant contribution of this method when we plotted the learning curves for each model. For the majority of the best-performing models, the learning curves displayed high fluctuation, which indicated the model's unsuitability for generalisation. However, the learning curves of two models that were trained on 100,000 reduced images exhibited smooth lines by a large margin, namely (a) the model that was trained with images of the proposed colouring method and processed with $\alpha = 0.6$, and (b) the model that was trained with images of [17]'s colouring method and processed with $\alpha = 0.3$. Of the two models, the first one showed the smoothest curves, which can be seen

in Figure 8. For this purpose, and since all numerical results exhibited comparative values (between 93–94%), we picked the specifications of this method to form the building block of our IDS. As a result, our IDS was trained on 10,000 images of the proposed colouring scheme that had been processed with an $\alpha$-cut of 0.6. We summarise the hyperparameters of the winning model in Table 11.

**Table 4.** Performance metrics for [18], without $\alpha$-cut processing.

| | Colouring Method: [18], No Processing with $\alpha$-Cut | | | |
| --- | --- | --- | --- | --- |
| **Dataset Size** | **Accuracy (%)** | **Precision (%)** | **Recall (%)** | **F-Score (%)** |
| 1250 | 80.80 | 83.05 | 77.77 | 80.32 |
| 2500 | 89.60 | 87.21 | 92.80 | 89.92 |
| 5000 | 88.40 | 84.78 | 93.60 | 88.97 |
| 10,000 | 89.80 | 92.70 | 86.40 | 89.44 |

**Table 5.** Performance metrics for [17], without $\alpha$-cut processing.

| | Colouring Method: [17], No Processing with $\alpha$-Cut | | | |
| --- | --- | --- | --- | --- |
| **Dataset Size** | **Accuracy (%)** | **Precision (%)** | **Recall (%)** | **F-Score (%)** |
| 1250 | 84.80 | 95.83 | 73.01 | 82.88 |
| 2500 | 88.40 | 91.37 | 84.80 | 87.96 |
| 5000 | 92.60 | 95.31 | 89.60 | 92.37 |
| 10,000 | 94.50 | 97.23 | 91.60 | 94.33 |

**Table 6.** Performance metrics for [17], processed with $\alpha = 0.3$.

| | Colouring Method: [17], Reduced for $\alpha = 0.3$ | | | |
| --- | --- | --- | --- | --- |
| **Dataset Size** | **Accuracy (%)** | **Precision (%)** | **Recall (%)** | **F-Score (%)** |
| 1250 | 92.00 | 90.76 | 93.65 | 92.18 |
| 2500 | 89.60 | 93.04 | 85.60 | 89.16 |
| 5000 | 91.80 | 95.63 | 87.60 | 91.44 |
| 10,000 | **93.89** | **95.82** | **91.80** | **93.76** |

**Table 7.** Performance metrics for [17], processed with $\alpha = 0.6$.

| | Colouring Method: [17], Reduced for $\alpha = 0.6$ | | | |
| --- | --- | --- | --- | --- |
| **Dataset Size** | **Accuracy (%)** | **Precision (%)** | **Recall (%)** | **F-Score (%)** |
| 1250 | 87.20 | 94.33 | 79.36 | 86.20 |
| 2500 | 88.80 | 91.45 | 85.60 | 88.42 |
| 5000 | 89.60 | 93.04 | 85.60 | 89.16 |
| 10,000 | 92.80 | 95.72 | 89.60 | 92.56 |

**Table 8.** Performance metrics for our method, without $\alpha$-cut processing.

| | Colouring Method: Our Method, No Processing with $\alpha$-Cut | | | |
| --- | --- | --- | --- | --- |
| **Dataset Size** | **Accuracy (%)** | **Precision (%)** | **Recall (%)** | **F-Score (%)** |
| 1250 | 86.40 | 92.59 | 79.36 | 85.47 |
| 2500 | 90.40 | 94.69 | 85.60 | 89.91 |
| 5000 | 93.40 | 97.79 | 88.80 | 93.08 |
| 10,000 | 93.89 | 96.40 | 91.20 | 93.73 |

**Table 9.** Performance metrics for our method, processed with $\alpha = 0.3$.

| Colouring Method: Our Method, Reduced for $\alpha = 0.3$ | | | | |
|---|---|---|---|---|
| Dataset Size | Accuracy (%) | Precision (%) | Recall (%) | F-Score (%) |
| 1250 | 87.20 | 89.83 | 84.12 | 86.88 |
| 2500 | 89.20 | 95.37 | 82.39 | 88.41 |
| 5000 | 93.40 | 96.56 | 90.00 | 93.16 |
| 10,000 | 92.90 | 93.86 | 91.80 | 92.82 |

**Table 10.** Performance metrics for our method, processed with $\alpha = 0.6$.

| Colouring Method: Our Method, Reduced for $\alpha = 0.6$ | | | | |
|---|---|---|---|---|
| Dataset Size | Accuracy (%) | Precision (%) | Recall (%) | F-Score (%) |
| 1250 | 83.20 | 88.88 | 76.19 | 82.05 |
| 2500 | 91.20 | 94.78 | 87.20 | 90.83 |
| 5000 | 93.20 | 97.36 | 88.80 | 92.88 |
| 10,000 | **93.60** | **94.48** | **92.60** | **93.53** |

**Table 11.** Hyperparameters of the best performing ResNet50.

| Hyperparameters | |
|---|---|
| Batch | 32 |
| Optimiser | Stochastic Gradient Descent |
| Learning Rate | 0.1 |

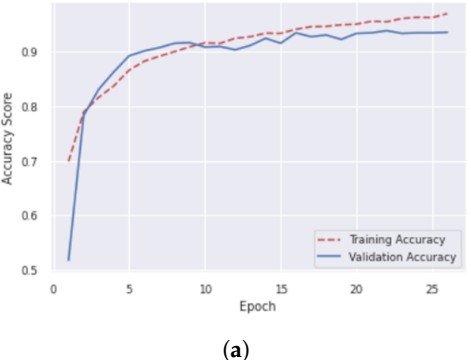 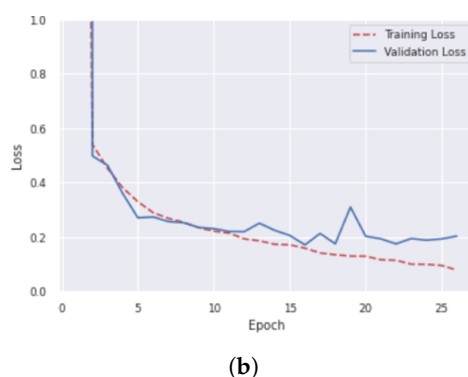

(**a**)        (**b**)

**Figure 8.** The learning curves of mean training and validation accuracies (**a**) and mean training and validation losses (**b**) during training of the best ResNet50 model. The model was trained on 10,000 images of the proposed colouring scheme that had been processed with an $\alpha$-cut of 0.6.

## 5. Discussion

The aim of the present research was to examine whether sparse representations of malware binary images carry sufficient information to become an integrated part of a DL-based IDS. As can be seen in the literature review, binary visualisation has exceeded the role of a malware analysis tool by evolving into an indispensable component of malware detection mechanisms, thanks to the widespread adoption of CNNs. At the same time, in an effort to ameliorate computational costs, recent innovations in NN models seek to experiment with the notion of sparsity; either internally, by pruning model components, e.g., relu, weight regularisation, etc. [96], or externally, e.g., by selecting an optimised storage format for sparse inputs [97], or by generally studying how input data dimensionality can influence prunability [98]. In fact, sparsity is expected to become an increasingly important issue in the future and even affect hardware choices [99].

To the best of our knowledge, no previous study (a) has used $\alpha$-cuts to create reduced forms of images or (b) has measured the efficacy of sparse representations in DL training.

Consequently, this is the first time this approach is being proposed for image-based malware detection. This can be considered a significant step forward in detection mechanisms that seek to operate in a real-time fashion, such as network traffic analysis. Indeed, one aspect of our method that compromised the accuracy of the model was the fact that, depending on its size, a single binary file would translate into *multiple* fixed-size images. However, this was compliant with the future goal of our work, which is to diverge from a host-based to a network-based IDS and focus on inside packet detection where unencrypted network flow will be represented as a continuous generation of images. At the same time, the disadvantage of generating multiple images per file imposed the need for careful ASCII table clustering to maximise pattern construction across the binary.

Another restriction imposed by our objective to prepare the ground for a network-based IDS was the need to concentrate strictly on binary classification, or simply perform malware detection. In other words, we had to test our methods solely on the distinction between malicious and benign activity and have this classification serve as a prototype for network traffic. For this reason, it was found unnecessary to perform malware family classification, as well-established differentiations between family labels, which could lead to forcedly high accuracy scores and would not depict realistic capturing of unknown attacks in a network flow environment. This differs from the majority of other studies that focus on family classification, which often perform classification with the help of image feature extraction. One such example can be found in the work of [100] who successfully predicted family class from the Malimg dataset. Additionally, as concluded in the survey of [101], malware classification that relies on feature extraction as per class can lead to redundant information and negatively affect accuracy rates while demanding constant updates. From the above analysis, it is evident that we concentrated our efforts on including as many factors as we could to simulate the restrictions of a realistic IDS.

In this regard, we measured the performance of several hyperparameterised ResNet50 models that were trained on image datasets. The datasets, which were created from files of malicious and benign code, differed in size, colouring method, and $\alpha$-cut processing. The colouring methods included the method of [18], the method of [17], and our proposed colour classes, while the datasets of the last two methods were also processed with $\alpha$-cuts. For each colour method and processing (or lack thereof) combination, we tested four different dataset sizes.

As we observed from the results, the performance metrics were positively affected as the dataset size increased. However, the colouring approach of [18] displayed limited improvement, and none of the experiments were able to reach an accuracy of 90% (Table 4). This proves [18]'s inefficiency to construct meaningful patterns of colourful pixels that could reveal the presence of malicious code and can be attributed to its limited number of colour classes.

In contrast, our proposed approach and the approach of [17] displayed consistently improving scores, exhibiting larger gaps between the scores of the smallest and the largest dataset sizes (Tables 5–10). Notably, the difference in results between the best-performing models of our proposed colouring scheme and the work of [17] were not numerically significant. Indeed, both methods acquired comparative accuracies and f-scores of around 93–94% and 92–94%, respectively.

It is worth mentioning that the difference between these two approaches (when not applying any $\alpha$-cutting, see Tables 5 and 8) lies in the number of colour classes (we proposed a more detailed clustering of the ASCII table), while both have incorporated an FIS that adjusts the pixel colour tone according to its surroundings. Nevertheless, applying $\alpha$-cuts had different effects on them visually, as the initial RGB values per pixel differ significantly.

Another interesting comparison regarding these two methods would be that for each approach, we observed a slight decline between the experiment that used the unprocessed dataset and the experiments that used $\alpha$-cut processed datasets (Tables 5–10). More precisely, the best-performing model of the unprocessed dataset of [17]'s approach (Table 5) surpassed the best-performing model of the reduced datasets (Tables 6 and 7). Similarly, the best-

performing model of the unprocessed dataset of our approach (Table 8) surpassed the best-performing model of the reduced datasets (Tables 9 and 10).

From these findings alone, it would seem that processing an image with an $\alpha$-cut removes valuable information and impedes the model from making successful predictions. However, when examining the learning curves of all experiments for these two methods, it was evident that the models would not be able to generalise well, despite high scores or not. More precisely, almost all graphs exhibited high volatility, indicating a lack of correct predictability for new instances. Surprisingly, the experiment of our proposed method, whose 10,000 image dataset was reduced by $\alpha = 0.6$, and the experiment of [17]'s method, whose 10,000 image dataset was reduced by $\alpha = 0.3$ provided solid indications for robust predictions: both models' curves exhibited steady increases for training and validation accuracy and steady decreases for training and validation loss, with the first model being the closest to the ideal. This is consistent with the literature cited by [102,103], who emphasise the effectiveness of an increasing dataset size even over algorithm selection. At the same time, given the curves' fluctuation, for the remaining experiments of [17]'s approach (Tables 5 and 7) and ours (Tables 7–9), a dataset of 10,000 images did not make any contributions towards model robustness. We should also note that the experiment with the approach of [18] for the 10,000 image dataset displayed good learning curves too. However, we had to reject it altogether due to lower accuracy scores.

Given the robustness of the ResNet50 model that incorporated our proposed colouring method and was processed with an $\alpha$-cut of 0.6 (for a fuzzy set of increasing linear shape), we are confident that with a larger dataset (of more than 10,000 images) and enough training, our evaluation metrics would report greater scores (>93.6% accuracy and >93.53% f-score). Undoubtedly, and despite the restrictions we imposed in our method, for the dataset of 10,000 images, $\alpha$-cuts provided model robustness where unprocessed datasets failed to do so. From the above findings, we can safely conclude that $\alpha$-cuts can be regarded as a valuable image processing method. As we observed, they not only contribute to matrix sparsity but can also assist in demanding image datasets of synthetic representations (as opposed to natural images, e.g., of objects, animals, etc.). At this point, it is important to note that each type of dataset requires experimentation with the value $\alpha$, fuzzy set shape, and dataset size to decide on the appropriate level of processing.

As mentioned earlier in this paper, a key strength of this processing method is that it is colour model-independent. $\alpha$-cuts can be applied to any type of image, as long as the colour model values defining each pixel are well-defined fuzzy sets. This is particularly important since there is a plethora of popular colour models involved in image processing. At the same time, several image segmentation methods are not as simple or possess disadvantages. For example, although the RGB model seems to serve image processing well, there are still major critiques of its application in colour segmentation [64,104,105]. Another important but not as straightforward contribution of this method is that reducing an image into a matrix of 0 s and 1 s can act as an extra layer of data anonymisation for the already converted image. This enhances compliance with GDPR's "privacy by design policy", which requests that user data be processed to a satisfactory state of anonymity as mentioned in Section 1.

Despite the novelty of these findings, a number of limitations need to be noted. First, we were only able to test our datasets for a limited number of $\alpha$-cuts. Our initial thoughts included testing for $\alpha = \{0.1, 0.2, \ldots, 0.9\}$; however, computational time and costs limited our experiments to two $\alpha$-cuts. However, it would be interesting to observe how different $\alpha$-cuts and large enough datasets contribute to model effectiveness. Second, we arbitrarily defined the red, green, and blue channels as linear-shaped fuzzy sets. This rather simple shape served as a straightforward method where we could easily observe the relationship between value $\alpha$ and visual colour change in images. More specifically, the higher the value of $\alpha$, the more 0 s the processed image would have ($^{\alpha}A$ would become smaller). Hence, for an increasing $\alpha$, images approached black. By contrast, this relationship would not be as straightforward when applying an $\alpha$-cut to a colour channel defined by a symmetrical triangular fuzzy set of $[2x, 2 - 2x]$. In that respect, geometrically-wise, changing $\alpha$ would

affect the relationship between median values of the interval $[0, 1]$ of the fuzzy set (which would turn to 1 s) and right/left boundaries (which would turn to 0 s). In any case, the rate at which an image turns darker is related to the initial RGB values (the colours of the specific dataset).

Another limitation of our study was the fact that we evaluated our proposed image processing technique on a synthetic dataset. In our case, this meant that we were not intentionally detecting a specific pattern or object with defined edges and characteristic contrast from its pixelated environment. Hence, our method was tested on visually unspecific patterns, something that would undermine the testing scores, whereas natural images show visually improved thresholding. Nonetheless, our findings on a challenging dataset provided satisfactory evidence for the robustness of the method. Finally, we should not fail to mention an inherent limitation that comes with space-filling curves. It is true that, despite their locality preservation property, a small change in the sequence of the input data (e.g., a small shift of the binary's ASCII characters) will produce a different pattern of pixels. Undoubtedly, this is another drawback that adds to the existing complication of splitting a file into multiple images.

## 6. Conclusions

In this study, we employed binary visualisation on malware and benign files to perform malware detection for the construction of a CNN-based IDS. However, the novelty of this paper lies in the image conversion process, where we introduced the application of $\alpha$-cuts to the RGB pixel values of the images along with a new colour grouping of the ASCII table. More specifically, the $\alpha$-cuts were implemented as a way to achieve a sparse representation of the images. Evidently, the higher the value of $\alpha$, the greater the sparsity of the matrix.

We tested our method on a variety of dataset sizes and evaluated its effectiveness according to the detection rates of hyperparameterised ResNet50 models. At the same time, we acknowledge that our study presented some limitations, such as multiple image splitting per file, experimentation with a small number of $\alpha$-cuts and fuzzy set shapes, as well as the lack of defined patterns and objects that natural images would provide. However, as reported by our results, we were able to acquire two robust high-score models that were trained on $\alpha$-cut-processed images where other models failed. To the best of our knowledge, this is the first time that $\alpha$-cuts have been used in image processing. Consequently, this provides an important contribution to image processing and thresholding in real-time practices such as IDS operations.

In the future, we plan to use these results to concentrate our efforts towards the development of a network-based IDS while expanding our experiments by implementing a larger number of $\alpha$-cuts and possibly defining a different shape for the fuzzy set of the value colour. At the same time, we plan to test our image processing methodology on a more straightforward dataset such as natural images in order to fully confirm our method's effectiveness.

**Author Contributions:** Conceptualization, B.S. and S.S.; methodology, B.S., I.M., S.S. and B.P.; software, I.M. and B.S.; validation, S.S., B.S. and B.P.; formal analysis, B.S. and I.M.; investigation, B.S. and I.M.; resources, S.S. and B.S.; data curation, I.M. and B.S.; writing—original draft preparation, B.S.; writing—review and editing, B.S., S.S. and B.P.; visualization, B.S.; supervision, S.S. and B.P.; project administration, S.S. and B.P. All authors have read and agreed to the published version of the manuscript as well as the authors order.

**Funding:** This project received funding from the European Union's Horizon 2020 research and innovation programme under grant agreements 957406 and 101021936. The work reflects the authors' view and the Agency is not responsible for any use that could be made from the information it contains.

**Data Availability Statement:** Not applicable.

**Conflicts of Interest:** The authors declare no conflict of interest.

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
