# Peer review of "Image-Based Malware Detection Using α-Cuts and Binary Visualisation"

_applsci, doi:10.3390/app13074624_

Round 1

Reviewer 1 Report

The authors present a research on visualisation-based malware detection. They propose an original variation based on image conversion using alpha-cuts. The paper is well written and well organized. The research methodolohy is appropiate and results are solid.

For further works, it would be interesting to explain, how does the process of generating several images (I assume 256x256) from a single file affect (if so) the process?

Author Response

Many thanks for your review. Please see attached file for our replies.

Reviewer 2 Report

The authors propose a novel method that enhances color and pattern segmentation and achieves a sparse representation of the images. The authors considered each pixel's R, G, and B color values as fuzzy sets. 

The paper is well-written, significant content, and could be considered for publication.

My only remark is regarding experimental results. The authors should justify the choice of ResNet50 and explain it clearly. 

Author Response

(The authors gave the same response as above.)

Reviewer 3 Report

In this paper, authors present the  newest additions in the visualisation-based malware detection domain. The aim of the research is to examine whether sparse representations of malware binary images carry sufficient information to become an integrated part of a DL-based Intrusion Detection Systems (IDS). Authors used α-cuts to create reduced forms of images and has measured the efficacy of sparse representations in DL training.In this study, authors employed binary visualisation on malware and benign files to perform malware detection for the construction of a CNN-based IDS. However, the novelty of this paper lies in the image conversion process, where authors introduced the application of α-cuts to the RGB pixel values of the images along with a new colour grouping of the ASCII table. More specifically, the α-cuts were implemented as a way to achieve a sparse representation of the images. Evidently, the higher the value of α, the greater the sparsity of the matrix. authors tested their method on a variety of dataset sizes and evaluated its effectiveness according to the detection rates of hyperparameterised ResNet50 models.

1. Please ensure that all variables/symbols introduced in the manuscript are properly explained and the index of each symbol is correct and consistent in order to avoid confusion.

2. Please include information regarding the APIs, libraries and packages used for this project with their respective version numbers. Additionally, any details regarding the hardware used for those experiments are welcome. Information that could help reproduce and further understand this study on a programming level would be beneficial for future research and experiments.

3.  The manuscript is interesting, however, some amendments are required. The related work section should be improved by better discussing the limitations of the existing methods and discussing more related works.

 Several references about Malware Detection model can be added [1][2].

[1] Catal, C.; Gunduz, H.; Ozcan, A. Malware Detection Based on Graph Attention Networks for Intelligent Transportation Systems. Electronics 2021, 10, 2534. https://doi.org/10.3390/electronics10202534

[2] Kamran Shaukat, Suhuai Luo, Vijay Varadharajan, A novel deep learning-based approach for malware detection, Engineering Applications of Artificial Intelligence, Volume 122, 2023,106030, ISSN 0952-1976, https://doi.org/10.1016/j.engappai.2023.106030.

4. It is not clear why the authors selected this binary visualisation on malware method architecture, why not other architectures. The authors should justify it by discussing some references or experimentally 

Author Response

(The authors gave the same response as above.)
